# A Luminous Efficiency-Enhanced Laser Lighting Device with a Micro-Angle Tunable Filter to Recycle Unconverted Blue Laser Rays

**DOI:** 10.3390/mi12101144

**Published:** 2021-09-23

**Authors:** Xinrui Ding, Ruixiang Qian, Liang Xu, Zongtao Li, Jiasheng Li, Caiman Yan, Binhai Yu

**Affiliations:** 1National & Local Joint Engineering Research Center of Semiconductor Display and Optical Communication Devices, South China University of Technology, Guangzhou 510640, China; dingxr@scut.edu.cn (X.D.); rxiang_q@163.com (R.Q.); meztli@scut.edu.cn (Z.L.); jiasli@foxmail.com (J.L.); chamenyan@163.com (C.Y.); 2Institute of Semiconductor Science and Technology, South China Normal University, 55 Zhongshan Avenue, Tianhe District, Guangzhou 510631, China; xuliang@nationstar.com; 3Guangdong Provincial Key Laboratory of Semiconductor Micro Display, Foshan Nationstar Optoelectronics Company Ltd., Foshan 528000, China

**Keywords:** hybrid type laser lighting, micro-angle tunable filter, heat dissipation

## Abstract

In this work, a phosphor converter with small thickness and low concentration, based on a micro-angle tunable tilted filter (ATFPC), was proposed for hybrid-type laser lighting devices to solve the problem of silicone phosphor converters’ carbonizing under high-energy density. Taking advantage of the filter and the scattering characteristics of microphosphors, two luminous areas are generated on the converter. Compared with conventional phosphor converters (CPCs), the lighting effects of ATFPCs are adjustable using tilt angles. When the tilt angle of the micro filter is 20°, the luminous flux of the ATPFCs is increased by 11.5% at the same concentration; the maximum temperature (MT) of ATFPCs is reduced by 22.8% under the same luminous flux and the same correlated color temperature (CCT) 6500 K. This new type of lighting device provides an alternative way to improve the luminous flux and heat dissipation of laser lighting.

## 1. Introduction

Because of its excellent energy density, slope efficiency and brightness, laser light is in wide use in headlights, high resolution display [1,2,3,4]. For example, in high-resolution, intelligent headlights, spatial light modulators such as digital micro-mirror device (DMD) or liquid crystal display (LCD) provide enough high luminous flux to ensure the brightness of certain pixels. Such laser lighting systems have great potential [5,6], and researchers are committed to the continuous development of laser sources and phosphor converters. 

To package and protect the phosphor, it is proportionately dispersed in the matrix [7], which has phosphor powder bonding material that includes polymers (e.g., epoxy resin and silicone) and inorganic materials (e.g., glass, ceramics, and single crystals) [8,9]. Based on these materials, considerable research has been carried out on improving luminous flux, uniformity of the correlated color temperature and temperature stability [10,11,12,13,14,15].

To improve the luminous flux or extraction efficiency of a solid-state lighting system, some researchers in the fields of laser and light emitting diode (LED) lighting are committed to reducing total internal reflection loss and Fresnel reflection loss by structured surfaces or microstructures. Ding et al. [16,17] processed roughened V-grooves with controllable shapes, angles and positions on chip-on-board substrates and packaging layers. Compared with traditional chip-on-board structures, those with V-grooved structures had an increased photoluminescence efficiency (PE) of 31.9%. Pan et al. [18] analyzed the influence of pyramid microstructures on the optical efficiency of luminous elements and found that, after optimizing the density and angle of the pyramids, the PE of the chip increased from 9.04 to 17.72%. Similar strategies were also used on converters [19,20]. In addition to the structured surface and microstructures, some optical films or coatings were used to enhance the use of exciting rays to improve the PE in the previous works. Tang et al. [21] applied dichroic filters to the upper and lower surfaces of polymer-based phosphor converters. The sandwich structure with two filters and a phosphor converter enhanced the recycling of blue light and reduced its reflection loss, thus improving the PE to 180%. Oh et al. [22] also proposed a similar sandwich structure with a filter, a phosphor converter and a polarizer. In this way, the luminous flux of the phosphor converters with this structure was increased to 139%, and the correlated color temperature (CCT) consistency was better.

Although previous research indicated that structures or films on converters function well for improving PE, the scattering ability of micro-nanophosphor particles was also investigated. Fujita et al. [23] investigated the influence of the size of phosphor particles on luminous efficacy in Ce: Y_3_Al_5_O_12_–Ce^3+^ (YAG) phosphor-in-ceramics phosphor. They found that the luminous efficacy increased with particle size for the lowering scatter coefficient. According to Mie scatter theory, the cross section of microparticles in a unit volume decides the scattering coefficient [23,24,25]. In another aspects, Yang et al. [26] used a high density phosphor to increase center-light intensity. The scatter coefficient of phosphor with higher density was stronger compared with that of a lower density; thus, the input laser was trapped in the center of the phosphor. Zhou et al. [27] investigated the effects of particle size and density on optical performance of phosphor converters, and their results showed that a high concentration and low particle size contributed to large scatter coefficient. The work above reveals that particle size and density have a significant effect on phosphor converters and that the scatter characteristics are also a key parameter for the optical performance of phosphor converters.

The configurations for the high PE and luminous flux mentioned above also engender heat accumulation because the more luminous flux that is generated the more heat is transformed due to nonideal quantum efficiency. Therefore, laser lighting still faces the problem of heat dissipation under high-power density. For the laser light source, a variety of heat dissipation methods have been developed, including microchannels, heat pipes, forced air cooling, and double heat sinks [28,29,30,31,32]. Though, at present, there are reliable means to dissipate the heat of a laser light, phosphor converters are still facing the challenge of heat dissipation. Trivellin et al. [33] studied the failure of laser-lighting base materials and found that when the power of the irradiated point exceeded the power intensity limit (550 mW/0.1145 mm^2^), the material carbonized. Ding et al. [34] found that, though an Al substrate was used for heat dissipation, the phosphor converters with high concentration and large thickness still carbonized on the upper surface. To reduce heat accumulation, Narendran et al. [35] proposed dispersing the phosphors in the heat conduction matrix to form a regional phosphor structure, but its ultimate thermal conductivity was limited to that of the polymer materials. Ma et al. [36] proposed reducing concentration and thickness to improve temperature stability based on the study of temperature distribution in the polymer-based phosphor converters, but this led to a decrease in optical efficiency. Materials with high thermal conductivity and stability, such as glass and ceramics, contributed to improving thermal performance [37,38]. However, high fabrication costs and brittleness limit their application. Nevertheless, it is an alternative way to use a silicone converter with good heat dissipatio4n and optical performance for laser lighting device.

In addition, to the best of our knowledge, once the converter is manufactured, its lighting effects, such as CCT, luminous flux, cannot be changed. Therefore, based on the above-mentioned research, mainly in improving thermal stability and luminous flux, a new type of lighting effects-adjustable and hybrid lighting system with enhanced luminous flux and heat dissipation using a micro tunable angle filter is proposed. In this work, the filter separates blue and yellow light to reduce yellow reabsorption loss and reuse residual blue light. Experimental results showed that the lighting effects of the micro-angle tunable tilted filters (ATFPCs) are adjustable using the filter tilt angles, and the luminous flux and the heat dissipation are enhanced.

## 2. Materials and Methods

### 2.1. Characteristics of the Optical Filters

The optical filters were manufactured by Shenzhen Giai Photonics Co., Ltd. (Shenzhen, Guangdong province, China) The filter is reflective, which allows long-wavelength light to pass through and reflects short-wavelength light, as shown in Figure 1a. For a filter, its cut-off frequency is bule-shifted when incident angle increases. The cut-off frequency of 545 THz (wavelength: 550 nm) guarantees that light with a lower wavelength will always be reflected when the incident angle increases, so a 550 nm filter was adopted. The characteristics of its transmission and reflection curves are shown in Figure 1b. Filter reflectance of light with a wavelength of less than 550 nm is greater than 92%, and the transmittance of light with a wavelength of more than 550 nm is greater than 92%.

### 2.2. Fabrication of YAG Yellow Phosphor Converters

The Y_3_Al_5_O_12_–Ce^3+^ (YAG) yellow phosphor was manufactured by Shenzhen Zhangwanglong Technology Co. Ltd. (Shenzhen, Guangdong province, China). Polydimethylsiloxane (PDMS, manufactured by Shanghai Aladdin Bio-Chem Technology Co., Ltd (Shanghai, China) was used to encapsulate the phosphor particles. First, PDMS A and crosslink agent B were mixed in a mass ration of 10:1. Second, the phosphor was added to prepare a mixture of 10% phosphor. Third, the mixture was stirred in a vacuum defoamer for about 5 min. Fourth, the mixture of phosphor and PDMS was injected into premade modules with a thickness between 0.25 and 2 mm, with an interval of 0.25 mm. Finally, the samples were kept in an oven at 90 °C for 30 min for the matrix to solidify completely. The mass friction of the second step was adjusted from 10 to 30% with an interval of 2.5%. The four steps above were repeated to prepare phosphor converters with different concentration of YAG: Ce^3+^.

### 2.3. Experiment Setups

As shown in Figure 2c, the ATFPC set up consisted of four parts—a PDMS-based phosphor converter, square filter, phosphor converter fixture with a hole and filter fixture with an angle scale that rotated around its own central axis as shown in Figure 2a and Figure 3. The rotation angle of the filter fixture could be precisely controlled. The phosphor converter was tied to, and fixed on, the converter fixture and covered the hole. The two fit closely. The phosphor light passed through the hole, and the blue light and part of phosphor were reflected back to the phosphor layer; thus, a hybrid-type lighting system was formed by the transmission characteristics. The filter was fixed on the angle-tunable optical fixture, the central axis of which coincided with the rotation axis of the latter; thus, the filter was angle tunable when the fixture rotated. In each experiment, the height of the fixture axis and distance between the hole and converters remained unchanged. As a reference, a phosphor conversion system was also set up with a phosphor converter and two fixtures with no filter as shown in Figure 2b,d. All the elements of the ATFPCs were fixed on a small optical platform as shown in Figure 3.

The spectra of different laser-driven phosphor converters were experimentally investigated in a 900 mm diameter integrating sphere. All the phosphor systems were placed in it apart from the laser source which input the laser into the integrating sphere through an optical fiber. An optical detector was linked to a spectrometer from Ocean Insight (Dunedin, FL, USA). The thermal performance test of the ATFPC and CPC samples were carried out at room temperature by an infrared (IR) camera from FLIR Systems Inc., (Boston, MA, USA). The IR camera focused on the front surface to measure the temperature of the phosphor converters. In addition, the images were obtained by Zeiss’s MERLIN scanning electron microscope (SEM) (Jena, Thuringia, Germany). The scattering characteristics of the phosphor particles was obtained through a spatial photometer, which had a measurement range from −90 to 90°.

## 3. Results and Discussion

### 3.1. Scatter Characteristics of Phosphor Converters

The SEM images of the phosphor particles were first explored to determine their sizes and morphology. Figure 4a is the images of phosphor particles at different magnifications. It shows irregular morphology with a mean particle size of 15 μm. Correspondingly, we obtained the scattering characteristics of these particles with different concentration and thickness using the spectrometer. In Figure 4b, the normalized intensity of scattered light by the phosphor converters tended to be smooth with increasing concentration from 17.5 to 30%. A large light intensity still remains in the center positions. Since the scattering characteristics of particles partly depend on the concentration of the particles [23,24,25,26,27], greater concentration means a more uniform luminous intensity distribution. The variation of scattering characteristics with concentration is helpful for analyzing the gain effect of the micro-angle tunable tilted filter.

### 3.2. Effects of Tilt Angles on Optical Performance

Considering the influence of angles on the luminous flux and the CCT of the ATFPCs, the concentration was fixed at 10% and the thickness at 0.5 mm to ensure a single variable. The power density of the laser was 10 W/cm^2^, below the power density limit in [33]. Figure 5a shows the spectra of the ATFPCs with different tilt angles and CPCs. In the yellow light band (480–780 nm), the intensity of the ATFPCs rises in the range of 0–20° and then falls, which is always higher than that of the CPCs. In the blue light band (380–480 nm), the intensity of the CPCs was the highest. Then the spectra and sub-spectra were integrated to obtain the total luminous flux and that of the residual blue and yellow light as shown in Figure 5b,c. In Figure 5b, the total luminous flux of the ATFPCs is superior to that of the CPCs. The relative increase in luminous flux of ATFPCs over CPCs kept rising in the range of 0–20° and reached maximum value 11.5% at 20°. From 20 to 30°, although the luminous flux of ATFPCs decreased gradually, it still showed advantages over the CPCs. Figure 5c shows that the flux of residual ATFPC blue light decreased to 36.3% and the flux of yellow light increased to 207.4% compared with the CPCs. It is worth noting that the CCT of the phosphor converters increased from 6200 to 7100 K and it reached 6500 K at 20°, which was close to the standard white light source, as shown in Figure 5d. The CCT and luminous flux changed with the filter tilt angle, indicating that the lighting effects of AFTPCs were tunable. Furthermore, the luminous flux of the ATFPCs with tilt 20° angle was the highest. The CCTs were also the closest to that of standard white source. The corresponding International Commission on illumination (CIE) chromaticity coordinates is presented as Table 1 and Figure 6a in detail. The title “WTO” in Table 1 means a CPC without a tilt filter.

The increase in luminous flux resulted from the reuse of blue light excitation and is determined by the tilt angle α. When the concentration of the phosphor converters is only 10%, the phosphor particles in the directly illuminated area of high energy density laser are saturated, as shown in Figure 2d. Even if the remaining excited light returned along the original path to the emitting area, the enhancement of the light flux was limited. When the angle increased, a new excitation region was separated from the original one, which excited more phosphors in the unsaturated or unexcited state as shown in Figure 2c. Therefore, with the increasing angle, the effects were gradually enhanced. However, the distance between the two emitting regions positively correlated with the tangent value of the filter inclination. When the angle increased to a certain value, the residual blue light was too dispersed to shine completely on the phosphor converters, leading to increased residual blue light intensity. Then the luminous flux decreased and the CCT kept rising. It showed that the lighting effects of the ATFPCs could be adjusted by adjusting the inclination of the filter.

### 3.3. Effects of Concentration on Optical Performance

It is already known that the lighting effects are tunable using the tilt angle and that the effects are the best at 20° (Figure 5). To further the optical and thermal performance of the ATFPCs optimize, the tilt angle 20° was chosen in this section to continue the optimization. The laser power density was set to 10 W/cm^2^. The thickness of converters was 0.5 mm. Figure 7 discusses the effects of the phosphor converter concentration on the enhancement of the luminous flux and the CCT. The advantages of the ATFPCs over the CPCs in lighting effects were also analyzed. As shown in Figure 7a,b, the spectra of the ATFPCs and the CPCs indicated that the intensity in blue light band decreases with increasing concentration. But in yellow light band, the trend was just the opposite. By integrating the spectra, the luminous flux of the CPCs and the ATFPCs was obtained in Figure 7c. The results showed that the total flux of the CPCs increased with increasing concentration, and ATFPCs were similar. Moreover, the luminous flux of ATFPCs with 10% phosphor concentration was 98.2% of the CPC flux with 20% concentration. Therefore, with the ATFPC configuration, only half of the phosphor dose achieved the same luminous flux, which saves precious rare earth phosphors. But when the concentration rose to 30%, the luminous flux of the ATFPCs was lower than that of the CPCs with the same concentration. Therefore, the dominant range of ATFPC concentration was 10–27.5%. The increase in ATFPCs over CPCs also decreased with increasing concentration—from 16.4% with 10% concentration to −5% with 30% concentration as shown in Figure 7c. Figure 7 indicates that, for the CPCs, if it was required to reach standard white light of about 6500 K, the phosphor concentration needed to be increased to 25%. In contrast, the ATPFCs achieved that with only 10% phosphor concentration. This proved that, compared to CPCs, ATPFC required a lower phosphor concentration, which benefited from the improved filter light conversion. The corresponding CIE chromaticity coordinates are presented as Table 2 and Table 3, and Figure 6b,c. They show the linear relationship between the CIE chromaticity coordinates of ATFPCs and CPCs with different concentrations.

When the phosphor concentration rose, the phosphor particles had a strong scattering ability to the laser, as shown in Figure 4b. After the first time the blue laser penetrated the phosphor converters, the residual laser had been fully scattered and the central light intensity was quite low. Therefore, the filter in the ATFPCs had a problem reguiding all the residual blue light back to the converters. In addition, the filter absorbed the blue and yellow light to a certain extent. Finally, the gain brought by the reuse of blue light by the filter was smaller than the loss caused by its absorption. Thus, ATFPCs performed than CPCs in luminous flux with high concentration. Instead ATFCs had more significant advantages in the low-concentration fluorescence range.

### 3.4. Thermal Performance of ATFPCs

It is already known from the above analysis that ATFPCs with 10% concentration performed better both in luminous flux and the CCT compared with the CPCs whether with the same concentration (10%) or double (20%). The heat dissipation of the ATFPCs needed to be verified to explore further the application potential of the ATFPCs. The concentration of the ATFPCs was 10%, that of the CPCs 10% (denoted as CPC #1) and 20% (denoted as CPC #2). In this experiment, the laser power density was 10 W/cm^2^, and the thickness of the ATFPCs and CPCs was 0.5 mm. Figure 8 shows the difference in temperature distribution of the ATFPCs over CPC #1 versus tilt angle α. Here, heat dissipation is characterized by maximum temperature (MT). As shown in Figure 8, the maximum surface ATFPC temperature decreased by 11.7% from 77.7 to 69.5 °C with an increasing tilt angle. Compared with CPC #1, the increase in the maximum temperature of the ATFPCs was up to 15.9% at 0°, and as low as 1.6% at 30°. The temperature increase in the ATFPCs was caused by the secondary use of the laser. When the inclination angle was 20°, the MT of the ATFPC was 7.2% higher than that of CPC #1. Meanwhile, the PE of the ATFPCs improved 11.5% compared to that of the CPCs as shown in Figure 8f, which verified the above hypothesis.

Furthermore, the temperature distribution of the ATFPC with a 20° tilted angle was compared with the CPC under the same luminous flux. Therefore, CPC #2 with the luminous flux as large as ATFPC’s was also analyzed in Figure 9. The MT of the ATFPC and the CPC was 67 and 93 °C, respectively. Although the PE of the two was similar, the ATFPC temperature fell by 21.2 °C, which is 22.8% lower. This showed that our proposed strategy had excellent heat dissipation performance under the same luminous flux.

The high concentration and large thickness of the phosphor converters absorbed and transformed the blue light in the direct irradiation area and its adjacent area during the laser first incidence, but it was accompanied with a lot of heat generation. At the same time, the thermal conductivity of the polymer-based phosphor converters was poor, resulting in heat accumulation in a small area, and finally a carbonized surface. Low-concentration and small-thickness phosphor converters can reduce the local MT by reducing the conversion and absorption of the incident blue light, but it also means low luminous flux and high CCT. This thorny contradiction can be effectively resolved through the proposed ATFPC strategy. When such converters are equipped with a micro-angle tunable tilted filter, the filter reflects the residual blue light passing through the ATFPCs in another direction, so that the high-intensity blue light energy can be dispersed, thereby improving the heat dissipation of the ATFPCs.

### 3.5. Intensity Regulation of Uneven Lumination

To further determine and optimize the intensity distribution of ATFPCs, a simulation model of ATFPCs at 20° was set up and run by TracePro as shown in Figure 10d. The ATFPC was placed at 40 on the *z* axis. A parabolic reflector with a 40 focal length was placed at the coordinate origin. An object plane was placed at 200 on the *z* axis. Figure 10 indicates the real and simulated light intensity distribution of ATFPCs at a 20° tilt angle. In Figure 9a, there are two peaks of intensity distribution curves of ATFPCs at −50 and 20°, in the 0–180° semi-planar. The peak at 50° is the nonideal peak of the leaked bule light and the green dot curve is ideal with no leaked blue light. The saddle shaped light distribution is undesirable for lighting. Without the reflector, the simulation results of the module are shown in Figure 10b. It is consistent with the related true color images with 505.18 lm flux in Figure 10e. It shows that there are two peaks of the curve in the 0–180° semi-planar. The results were similar to those in Figure 10b, meaning that the simulation is reasonable. To couple the two peaks, the parabolic reflector was introduced. The curves in Figure 10c tended to be smoother and more suitable for lighting, especially for headlights. Its true color images with 508.35 lm flux are shown in Figure 10f.

The intensity distribution of ATFPCs was uneven due to the reguiding of the micro-angle tunable tilted filter. In the normal of the filter, the intensity was stronger than in the direction of incidence. Therefore, there were two peaks with different intensities in the two directions. When the parabolic reflector was introduced, the weak and strong peaks were gathered to the center and coupled to each other. Hence, the valley in two peaks disappeared and a new peak occurred at 10°, indicating that the reshaping of intensity distribution curves is easily done without extra luminous flux loss.

## 4. Conclusions

This work proposed a new type of adjustable laser lighting system with enhanced luminous flux and heat dissipation using a tunable angle to recycle laser. Optical and thermal performances and light distribution were experimentally or simultaneity investigated, respectively. Taking advantage of the tilted angle, the reflective filter reguided residual blue light to another region on the same phosphor converter after the first incidence and partial absorption; thus, two luminous regions were formed and the energy of the laser was reused. The results showed that the lighting effects of the ATFPCs, such as luminous flux and CCT, can be adjusted by tuning the tilt angle of the filter. The optimization of the spatial distribution of intensity was discussed as well. At 20°, the ATFPCs we proposed had 11.5% more luminous flux and increased photoluminescence efficiency compared with CPCs with the same concentration. Its CCT was closer to that of a standard white light source as well. Under the same luminous flux, though, the CCT of the two types of converters were all about 6500 K, and the concentration of the ATFPCs is just half of that of the CPCs. The maximum temperature of the ATFPCs was also 22.8% lower than for the CPCs. Based on the above conclusions, ATFPCs provide a new heat-dissipation solution for laser lighting systems, paving the way for their wide application.

## Figures and Tables

**Figure 1 micromachines-12-01144-f001:**
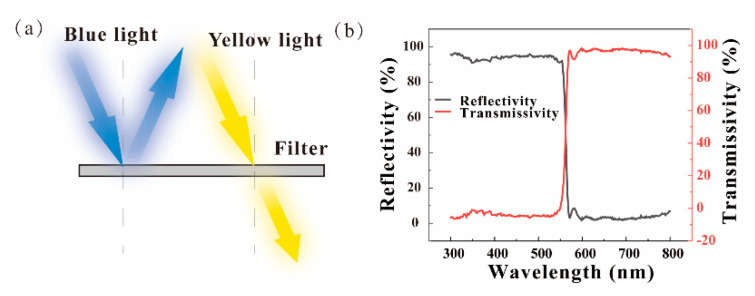
(**a**) Schematic diagram of the filter; (**b**) index of reflectance and transmission versus wavelength.

**Figure 2 micromachines-12-01144-f002:**
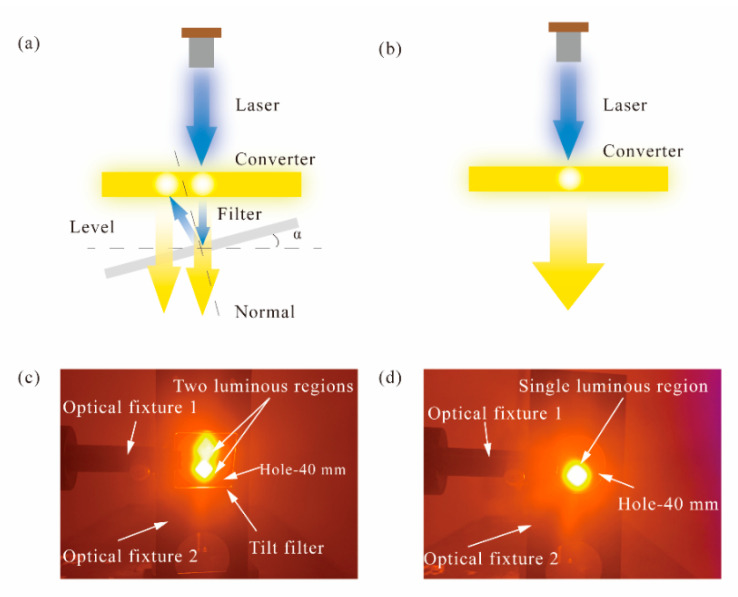
Diagrams of (**a**) micro-angle tunable tilted filters (ATFPCs) and (**b**)conventional phosphor converters (CPCs); actual lighting effect images of (**c**) CPCs and (**d**) ATFPCs under a brown filter to protect the camera.

**Figure 3 micromachines-12-01144-f003:**
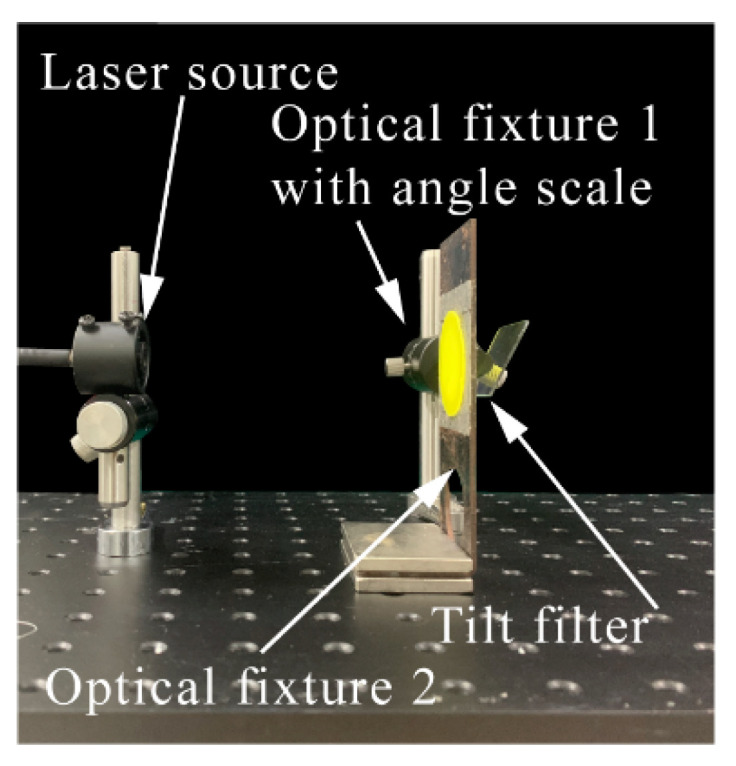
Configurations of the phosphor converters and actual lighting effects of the CPCs and ATFPCs.

**Figure 4 micromachines-12-01144-f004:**
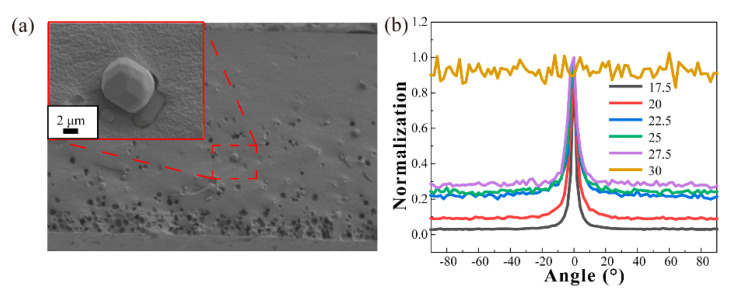
(**a**) The scanning electron microscope (SEM) images of microphosphor particles; (**b**) the scatter characteristics of phosphor converters with different concentrations.

**Figure 5 micromachines-12-01144-f005:**
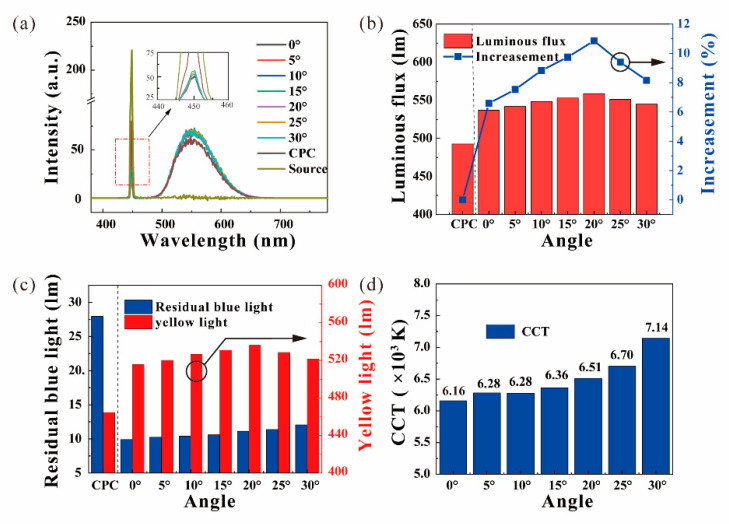
Effects of the title angle of the filter, spectra of (**a**) the CPCs, the ATFPCs; (**b**) luminous flux and increase in the ATFPCs over the CPCs; (**c**) residual bule rays and yellow light; (**d**) the correlated color temperature (CCT) changed with the title angle α.

**Figure 6 micromachines-12-01144-f006:**
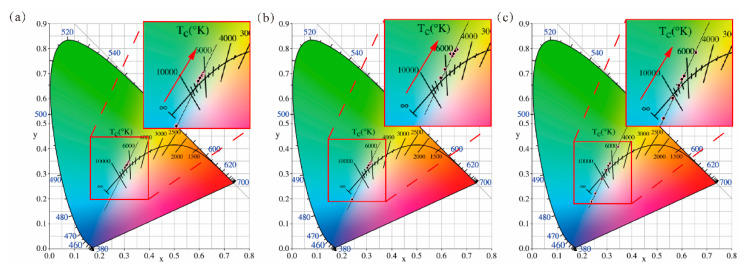
International Commission on illumination (CIE) chromaticity diagrams of (**a**) ATFPCs with different angles; (**b**) ATFPCs with different concentration; (**c**) CPCs with different concentration.

**Figure 7 micromachines-12-01144-f007:**
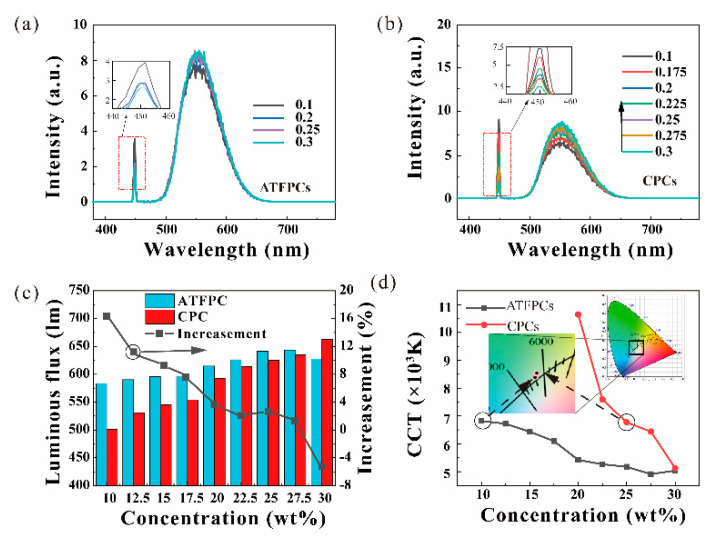
Effects of the concentration of phosphor particles, spectra of (**a**) CPCs and (**b**) ATFPCs; (**c**) luminous flux and relative increase in ATFPCs; (**d**) the CCT of ATFPCs and CPCs.

**Figure 8 micromachines-12-01144-f008:**
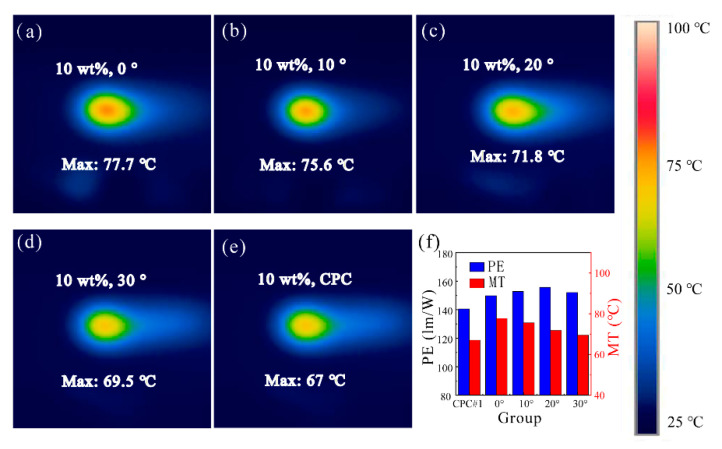
Temperature distribution of ATFPCs with (**a**) 10% concentration, 0°; (**b**) 10% concentration, 10°; (**c**) 10% concentration, 20°; (**d**) 10% concentration, 30° and the CPCs with (**e**) 10% concentration; and photoluminescence efficiency (PE) of (**f**) the ATFPCs as well as the CPCs, with the same thickness of 0.5 mm.

**Figure 9 micromachines-12-01144-f009:**
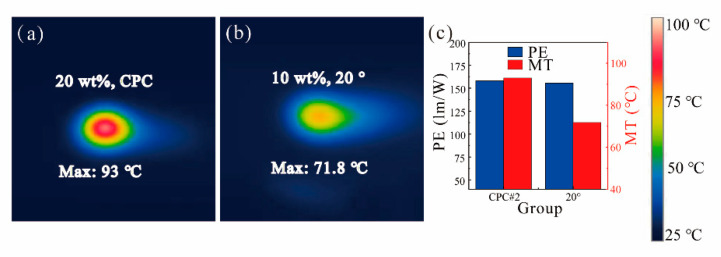
Temperature distribution of (**a**) the ATFPC; (**b**) the CPC; (**c**) PE of (**a**) the ATFPC; (**b**) the CPC.

**Figure 10 micromachines-12-01144-f010:**
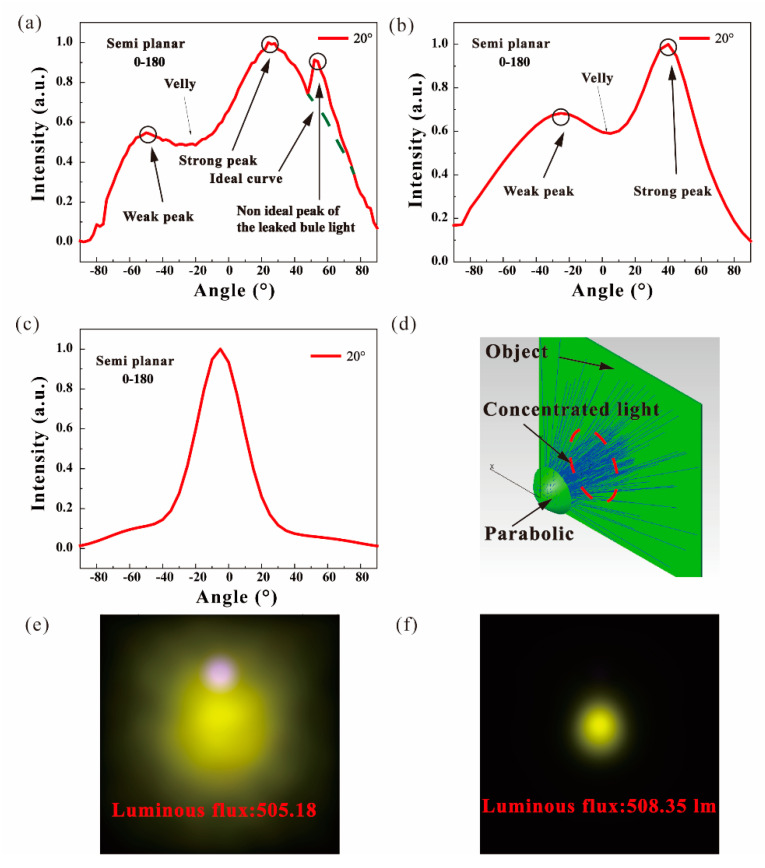
Light intensity distribution of (**a**) experiments, simulations (**b**) before and (**c**) after optimization; (**d**) simulation models of modulated ATFPCs, true color images of ATFPCS (**e**) without and (**f**) with a reflector.

**Table 1 micromachines-12-01144-t001:** CIE coordinates verse tilt angle.

CIE Coordinates	0°	5°	10°	15°	20°	25°	30°	WTO
x	0.3176	0.3154	0.3154	0.3138	0.3114	0.3083	0.3024	0.2419
y	0.3465	0.3428	0.3424	0.3413	0.3374	0.3334	0.3242	0.1973
z	0.3359	0.3418	0.3422	0.3449	0.3512	0.3582	0.3734	0.5609

**Table 2 micromachines-12-01144-t002:** CIE coordinates verse phosphor concentration of ATFPCs.

CIE Coordinates	10 wt%	17.5 wt%	20 wt%	22.5 wt%	25 wt%	27.5 wt%	30 wt%
x	0.3071	0.3185	0.3401	0.335	0.3433	0.3533	0.3486
y	0.3279	0.3522	0.3907	0.3821	0.3961	0.4083	0.4044
z	0.365	0.3294	0.2692	0.2829	0.2606	0.2383	0.2471

**Table 3 micromachines-12-01144-t003:** CIE coordinates verse phosphor concentration of CPCs.

CIE Coordinates	10 wt%	17.5 wt%	20 wt%	22.5 wt%	25 wt%	27.5 wt%	30 wt%
x	0.2381	0.2537	0.2979	0.2806	0.3074	0.3456	0.3123
y	0.1891	0.2213	0.3145	0.2787	0.3315	0.4075	0.3415
z	0.5728	0.525	0.3876	0.4408	0.3611	0.2469	0.3462

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
