# Peer review of "A Luminous Efficiency-Enhanced Laser Lighting Device with a Micro-Angle Tunable Filter to Recycle Unconverted Blue Laser Rays"

_micromachines, 2021, doi:10.3390/mi12101144_

Round 1

Reviewer 1 Report

The manuscript entitled “A luminous efficiency-enhanced laser lighting device with a micro angle tunable filter to recycle the unconverted blue laser rays” was interesting and well written by the authors. All the necessary measurements were done and the results were supported by the experimental data. However, throughout the manuscript, the authors mentioned CCT and luminous flux of the phosphor converter using a micro angle tunable tilted filter (ATFPC). But, I did not see any CIE chromaticity coordinates of the output light. If possible, please include the CIE coordinates for better understanding and it is more comfortable for readers if the authors present the results in the tabular form, for example, different concentrations and different temperatures, etc. In my view, the manuscript was suitable for its publication in this prestigious journal “Micromachines”.        

Author Response

Points 1: However, throughout the manuscript, the authors mentioned CCT and luminous flux of the phosphor converter using a micro angle tuneable tilted filter (ATFPC). But I did not see any CIE chromaticity coordinates of the output light. If possible, please include the CIE coordinates for better understanding and it is more comfortable for readers if the authors present the results in the tabular form, for example, different concentrations and different temperatures, etc.

Response 1: Thank you for the professional and valuable advice. In the revised version, the CIE chromatic coordinates of different configurations are listed as Table 1, Table 2, Table 3, for readers to understand better. The CIE chromaticity diagram was added as figure 7 as well.

More details are presetented in the attachment

Reviewer 2 Report

Manuscript reports on a new type of lighting effect-adjustable laser lighting system with enhanced luminous flux and heat dissipation based on proposed angle tunable tilted filter approach. The manuscript technological relevance is high to wLEDs and most probably will attract interested readers’ attention.

In our opinion, manuscript is well developed, with clearly defined objectives and methodology. However, manuscript weakest point is lack of generalization on the proposed approach which may expand applicability of the ATFPC if revised accordingly.

Specific comments:

Is there any dependence of the title angle on the phosphor plate surface roughness?

In Figure 5, there is seen different dependance of luminous flux, residual blue light intensity and CCT on the angle. What strategy or trade off approach authors propose to implement in real devices to address this issue?

It is recommended that, when considering the practical aspect of the proposed ATFPC technology, authors shall present discussion on filter mounting/tilt precision, stability of mechanical mounting parameters due to heat affecting blue light reflection angle.

Authors considered the light scattering characteristics of phosphors with different particle concentrations however; it would be beneficial to generalize on the effectiveness of the ATFPC when using phosphors with different particle size distributions what seems to be more realistic approach here. Will ATFPC work the same for phosphors with very small particles (e.g. quantum dots) vs. polycrystalline phosphors with large particles? Was this issue considered, and if so some discussion on Rayleigh vs. Mia scattering affected by title angle shall be provided.

Round 2

Reviewer 2 Report

Authors have addressed all reviewer’s concerns satisfactorily, thank you. Revised manuscript shall be accepted for publication in Micromaschines.